

# Enhanced analysis of large-scale news text data using the bidirectional-Kmeans-LSTM-CNN model

Qingxiang Zeng

College of Humanities and Media, Hubei University of Science and Technology, Xianning, Hubei, China

## ABSTRACT

Traditional methods may be inefficient when processing large-scale data in the field of text mining, often struggling to identify and cluster relevant information accurately and efficiently. Additionally, capturing nuanced sentiment and emotional context within news text is challenging with conventional techniques. To address these issues, this article introduces an improved bidirectional-Kmeans-long short-term memory network-convolutional neural network (BiK-LSTM-CNN) model that incorporates emotional semantic analysis for high-dimensional news text visual extraction and media hotspot mining. The BiK-LSTM-CNN model comprises four modules: news text preprocessing, news text clustering, sentiment semantic analysis, and the BiK-LSTM-CNN model itself. By combining these components, the model effectively identifies common features within the input data, clusters similar news articles, and accurately analyzes the emotional semantics of the text. This comprehensive approach enhances both the accuracy and efficiency of visual extraction and hotspot mining. Experimental results demonstrate that compared to models such as Transformer, AdvLSTM, and NewRNN, BiK-LSTM-CNN achieves improvements in macro accuracy by 0.50%, 0.91%, and 1.34%, respectively. Similarly, macro recall rates increase by 0.51%, 1.24%, and 1.26%, while macro F1 scores improve by 0.52%, 1.23%, and 1.92%. Additionally, the BiK-LSTM-CNN model shows significant improvements in time efficiency, further establishing its potential as a more effective approach for processing and analyzing large-scale text data

Corresponding author
Qingxiang Zeng,
18627091638@163.com

## INTRODUCTION

Amid the backdrop of today's information explosion era, research endeavors are underway concerning the extraction of visual elements from news text and the mining of media hotspots (*Norouzi, 2022*; *Ruan, Caballero & Juanatas, 2022*). The rapid evolution of the Internet and social media has ushered in both formidable challenges and promising opportunities for the media industry. Within the vast expanse of news data, conventional text processing methods often prove inadequate in effectively discerning crucial information and emerging hot topics. As such, the research aimed at visual extraction from news text and the identification of media hotspots holds immense significance and value.

Traditional text processing methods encounter significant challenges in effectively identifying key information and emerging hot topics. One of the primary limitations lies in their reliance on simplistic approaches such as keyword matching and statistical methods like Term Frequency-Inverse Document Frequency (TF-IDF). These methods lack contextual understanding, operating at the surface level by merely counting occurrences of words or phrases without interpreting their semantic meaning or relationships. As a result, they struggle to discern nuanced information and fail to capture the intricate dynamics of language usage that characterize complex topics and emerging trends. Moreover, traditional text processing techniques often grapple with the complexities of noisy and unstructured data. Textual data, rife with spelling errors, abbreviations, and domain-specific jargon, poses challenges for preprocessing and normalization, leading to potential inaccuracies in analysis outcomes. Additionally, scalability remains a significant concern as these methods were originally designed for smaller datasets and may falter when confronted with the vast amounts of text data generated daily. This limitation hinders their ability to provide real-time insights into evolving trends and timely identification of emerging topics.

By harnessing the power of visualization technology, topics, keywords, and prevailing trends within news texts can be comprehended more lucidly. This, in turn, empowers media professionals with a profound understanding of paramount issues and the hottest events that pique public interest. This understanding enables them to adapt the angles and strategies employed in their news reporting. Furthermore, the exploration of news text visual extraction and media hotspot mining can also provide pivotal support for decision-making processes. Within the media industry, the formulation of precise decisions hinges upon an accurate grasp of public needs and prevailing public sentiment (*Alanazi et al., 2022*). The extraction and analysis of visual data from news texts afford decision-makers a wellspring of valuable information and insights. It equips them with the ability to delve deeper into the currents of social and market trends, thereby facilitating more informed and astute decision-making.

In the current landscape, research on news text visual extraction and media hotspot mining has made notable headway. Researchers have devised a multitude of text visualization techniques aimed at presenting and dissecting the characteristics and structure of news text data. These methodologies serve to empower users with a clearer comprehension of textual information and facilitate the discovery of crucial details and trending topics. To automatically identify topics and keywords within texts, topic mining and keyword extraction methods are employed (*Zheng et al., 2023*). Furthermore, researchers leverage Natural Language Processing (NLP) tools and libraries, including NLTK, spaCy, Gensim, and more, to engage in text preprocessing, sentiment analysis, and semantic analysis (*Kochmar, 2022*). Simultaneously, data mining and machine learning methodologies such as clustering, topic modeling, and association rule mining are deployed to unearth valuable insights from the textual data (*Raza & Ding, 2022*).

Rule-based algorithms predominantly employ manual extraction of textual features for classification. Nevertheless, amidst the exponential surge in global big data reservoirs in recent years, coupled with the susceptibility of the classification method to the subjective cognition of annotators, the recourse to machine learning methods has become imperative

for the automatic labeling of text data features. Building upon conventional machine learning algorithms such as the Naive Bayes algorithm, SVM, and KNN, text representation predicated on these methods often grapples with issues of high dimensionality, sparsity, and incomplete extraction of feature information (*Hassan, Ahamed & Ahmad, 2022*). Presently, deep learning neural network models exhibit commendable efficacy in text classification, with commonly employed architectures including CNN (*Wan & Li, 2022*), LSTM (*Wang & Li, 2022*), and BiLSTM (*Zheng et al., 2022*). Nonetheless, these individual neural network models manifest suboptimal performance concerning the accuracy of text classification and feature extraction, particularly when confronted with news text data.

There are still some shortcomings in the research on news text visual extraction and media hotspot mining. First, data quality and reliability are an important issue. The source and quality of news data vary, which leads to lower prediction accuracy of models obtained on these training data. Secondly, the low computing speed caused by large-scale data is a thorny problem. Third, semantic understanding and topic modeling of text is still a complex problem and requires further research to improve accuracy and effectiveness. The contributions of this article are multifaceted and can be summarized as follows:

(1) Innovative methodology: This article introduces a novel and refined approach by melding the BiK-LSTM-CNN framework with emotional semantic analysis, facilitating the achievement of high-dimensional news text visual extraction and media hotspot mining. This unique amalgamation of techniques paves the way for a deeper understanding of textual data, unearthing latent insights that were previously concealed.

(2) Comprehensive BiK-LSTM-CNN architecture: The designed BiK-LSTM-CNN framework encompasses four integral modules: news text preprocessing, news text clustering, sentiment semantic analysis, and the core BiK-LSTM-CNN model. Each module contributes to the holistic process of information analysis and extraction, ensuring a comprehensive and thorough examination of the data.

(3) Empirical advancements: Through rigorous experimentation, this study presents compelling empirical evidence. Comparative analysis reveals that, when compared with the Transformer, AdvLSTM, and NewsRNN models, the BiK-LSTM-CNN demonstrates substantial enhancements.

The elucidation of the proposed method can be found in 'Methods', while 'Experiment And Analysis' unveils the principal results. Ultimately, the article concludes with insights into future directions and potential avenues of research, as expounded in 'Conclusions'.

## RELATED WORKS

Exploration into the visual extraction of information from news text and the mining of media hotspots stands as a prominent concern within the domains of natural language processing and data science. Through the adept analysis and comprehension of voluminous news text data, the extraction of meaningful information and its subsequent visual presentation serve as instrumental tools in enhancing the populace's comprehension of prevailing hotspots and trends within the media landscape. Current methodologies bifurcate into traditional approaches and those founded upon deep learning paradigms.

## Text representation

Text representation serves as a prerequisite for subsequent classification research. In the realm of real-life textual information, easily comprehensible by individuals but not directly recognizable by computers, there exists a need to transform it into numerical data through the intermediary stage of text representation, facilitating further computational processing. An influential factor impacting classification efficiency is the quality of text representation. The Bag of Words (BOW) model, introduced by *Harris (1981)*, holds a significant position in traditional feature extraction methods. It dissects documents within a document set into individual words, undergoes deduplication, and thereby obtains a lexical representation of the text. However, BOW suffers from a substantial drawback, namely, its inability to fully encapsulate contextual information within the text. Simultaneously, as the number of words in the dictionary increases, it may give rise to issues of data sparsity. In comparison to BOW, Latent Semantic Analysis (LSA) (*Dumais, 2004*) achieves document dimensionality reduction but falls short in explicating each dimension of the vector space, precluding a probabilistic understanding of the model.

*Mikolov et al. (2013)* proposed the Word2vec model, which, through training, yields word vectors that comprehensively consider contextual information, thereby significantly enhancing classification efficiency. While Word2vec provides high-quality word vectors, it still lacks effective means to aggregate them into high-quality document vectors. Consequently, *Le & Mikolov (2013)* extended Word2vec, introducing Doc2vec, capable of acquiring vector representations for sentences, paragraphs, and documents. In contrast to the Word2vec model, Doc2vec finds broader application in semantic mining problems. This is attributable to the structural addition of a Paragraph vector in the input layer of Doc2vec, which, in training, functions as a memory aid, considers the discourse sequence of the text, and effectively addresses inter-sentential correlation issues.

Text extraction models constitute a vital component of research on mining media hotspots. *Ao et al. (2020)* delineated a method utilizing the TextRank algorithm for news summarization and extraction of hot events. While the approach demonstrates practicality and efficacy, further refinement is requisite to accommodate diverse field-specific demands. Nevertheless, this method fails to establish temporal relationships between news texts. *Albalawi, Yeap & Benyoucef (2020)* proposed a news text visualization method based on topic modeling. Initially, the method employs a topic model to analyze news text, yielding thematic distributions for visual representation; this visualization aids readers in better comprehending content. To enhance the temporal efficiency of existing methods, *Tsann et al. (2021)* introduced a news hotspot visualization approach based on the TextRank algorithm, validating its feasibility and efficacy through experimentation. However, this method solely concentrates on the text itself, neglecting non-textual factors (such as time, location) and their impact on hotspot events. *Du et al. (2022)* primarily investigated news text analysis methods based on word vectors and topic models. Leveraging word embedding technology for feature extraction from news text, the text is transformed into vector representations. Subsequently, employing a topic model, the method models news text, yielding topic distributions for each document.

While the aforementioned studies adeptly identify and extract information from news texts, efficiency concerns arise when handling extensive datasets. A predominant focus of current research methods is short-text classification. Consequently, when handling long texts, the prevalent approach involves initially condensing them to resemble the length of short texts before classification. However, this practice may lead to a certain degree of information loss, thereby diminishing the effectiveness of classification.

## Text classification model

Convolutional neural network (CNN) and related fusion models exhibit advantages in news text feature extraction and summarization capabilities. Leveraging their strengths in integrating information, capturing temporal dynamics, and understanding contextual relationships, these models contribute to heightened accuracy and judicious decision-making. However, considerations regarding data availability, computational complexity, and generalization remain pivotal for their effective deployment in real-world scenarios.

In 2014, *Kim (2014)* introduced transformative adjustments to the input layer of CNN, presenting the TextCNN model for text classification. The model's essence lies in utilizing various convolutional kernels to extract local key information from the text but overlooks contextual semantic correlations. *Hochreiter & Schmidhuber (1997)*, addressing issues like gradient decay and explosion during training in RNN, along with the limitation of capturing information only in proximity to the current position, refined and optimized the RNN network structure. By controlling the temporal gating of neurons, they enabled the capture of both long and short-range dependencies in sequences. However, LSTM can only scan in one direction, failing to capture future or subsequent contextual semantic information. To comprehensively capture semantic information within sequences, *Schuster & Paliwal (1997)* continued to enhance the LSTM network structure, introducing BiLSTM, which concurrently scans the entire sequence both forward and backward using LSTM.

In single-label classification, whether employing traditional RNN or enhanced LSTM models, standalone text classification exhibits suboptimal accuracy. *Jin et al. (2020)* proposed an LSTM model based on multitask learning, concurrently addressing text classification and feature extraction. This model jointly considers sentiment classification and attribute extraction tasks, learning common and specific features for both tasks through the shared underlying LSTM model and separately trained top-level classifiers. *Zhao et al. (2020)* adopted the BiLSTM model, proficient in capturing temporal sequence information and contextual relationships within the text. Additionally, they introduced a novel timestamp embedding method, converting temporal information in the text into the input for the LSTM model to better capture temporal dynamics. Emphasizing attention mechanisms and multitask learning, *Khine & Aung (2020)* introduced a framework based on LSTM and self-attention mechanism for text classification and target sentiment analysis. *Alayba & Palade (2021)* proposed a method combining CNN-LSTM with rule-based approaches for sentiment analysis of microblog texts. Initially, this method transforms the text into high-dimensional vector representations using a pre-trained word embedding model.

Since 2023, significant advancements have been made in CNN and LSTM models for text classification, focusing on improving their efficiency, accuracy, and ability to handle more complex text data. Integrating advanced attention mechanisms with CNN and LSTM models has been a notable trend (*Khan et al., 2023*; *Cui et al., 2024*). Attention mechanisms allow models to focus on the most relevant parts of the input text, improving the interpretability and effectiveness of the model. Studies have shown that combining CNNs and LSTMs with attention layers can significantly enhance the model's ability to understand and classify text by emphasizing crucial words and phrases within the context. Recent studies have also explored the development of dynamic and adaptive CNN and LSTM architectures that can adjust their complexity based on the input data (*Hasib et al., 2023*; *Syed & Ahmed, 2023*). These models can dynamically select the most appropriate layers and parameters for processing different types of text, improving efficiency and reducing computational overhead. Adaptive models have shown promise in various text classification tasks, offering a balance between performance and resource utilization.In summary, CNNs struggle to effectively extract contextual dependencies within text, while LSTMs can only capture semantic information for the entire sentence, lacking the ability to capture local key features of the text. Addressing these challenges, this article proposes an enhanced BiK-LSTM-CNN combined with emotional semantic analysis for high-dimensional news text visual extraction and media hotspot mining. Through high-dimensional mapping and the generation of embedded vectors, the method utilizes a dual-layer BiLSTM to extract feature information from the input text, acquiring global contextual information. Subsequently, TextCNN is employed to extract deeper-level key information, followed by softmax for predicting the text's assigned category.

## METHODS

The BiK-LSTM-CNN model, employs a structured approach integrating four key modules: news text preprocessing, news text clustering, sentiment semantic analysis, and the core BiK-LSTM-CNN model. These modules collectively facilitate comprehensive data analysis and inspection by sequentially processing and refining textual data through distinct stages. The initial module, news text preprocessing, focuses on preparing raw news text data by standardizing its format and content. This involves tasks such as tokenization, removing stopwords, handling punctuation, and applying stemming or lemmatization techniques to normalize the text. By establishing a clean and consistent input format, preprocessing lays the groundwork for subsequent analytical stages, ensuring that the data is ready for more sophisticated processing.

Following preprocessing, the news text clustering module organizes the textual data into meaningful groups or clusters based on similarity. This clustering process categorizes articles or documents with similar content together, employing k-means clustering. By grouping related texts, clustering enhances the efficiency and focus of subsequent analyses within each cluster, facilitating targeted insights extraction.

The sentiment semantic analysis module then examines the sentiment expressed within each cluster of news texts. Using sentiment analysis techniques, it determines the polarity

and intensity of sentiments conveyed in the articles. This analysis provides a deeper understanding of public opinion, market sentiment, or social attitudes towards specific topics covered in the news, enriching the data with emotional and contextual insights.

At the core of the model lies the BiK-LSTM-CNN architecture, integrating BiLSTM and CNNs. This module processes the preprocessed, clustered, and sentiment-enriched news texts to extract complex semantic features and patterns. BiLSTM captures temporal dependencies and context over sequences of text, while CNNs detect local patterns and features within the textual data. Together, these components enable the model to learn sophisticated representations of the news articles, incorporating both temporal sequences and spatial structures.

## Framework

The news text extraction and media hotspot mining methodology posited in this article encompasses a pre-processing module, a text clustering module, an emotional semantic analysis module, and a BiK-LSTM-CNN feature learning module, delineated in Fig. 1. Specifically, the preprocessing module undertakes text refinement to eliminate noise inherent in the original data. Subsequently, the news text clustering module groups similar text information, enabling subsequent LSTM to glean information features from the text. The emotional semantic analysis module dissects the emotional semantics within news texts, forecasting potential hot information within the news. The primary objective of the BiK-LSTM-CNN module lies in feature extraction from news text and subsequent classification, culminating in the derivation of the final result.

## News text preprocessing

After obtaining the high-dimensional news text, this study first preprocesses the text to remove the noise in the original data. The preprocessing method is the TFIDF method (*Zhao et al., 2018*), as calculated by Eq. (1):

$$w_{i,j} = \frac{\frac{t_j}{\sum_{j=1}^{n} t_j} \times \log\left(\frac{N}{n_j} + 0,01\right)}{\sqrt{\sum_{i=1}^{N} \left(\frac{t_j}{\sum_{j=1}^{n} t_j}\right)^2 \times \left[\log\left(\frac{N}{n_j} + 0.01\right)\right]^2}} \tag{1}$$

where, $t$ is the frequency of word t appearing in the $M$-th document, $N$ is the number of all documents, and $n$ is the number of documents containing word $t$.

The utilization of the TFIDF algorithm facilitates the efficient filtration of ubiquitous terms; however, it is imperative to acknowledge notable shortcomings that warrant remediation. Notably, this algorithm exhibits substantial deficiencies. In its computation of contributions, it exclusively scrutinizes the impact of the frequency of feature words, neglecting an examination of the position and part of speech of these words (*Senbel, 2021*). Specifically, nouns and verbs inherently possess a more potent topical expression compared to adjectives, thus enabling a more precise articulation of document content. Furthermore, when words manifest within the title section, they inherently carry more significant meaning than their counterparts within the main text area.

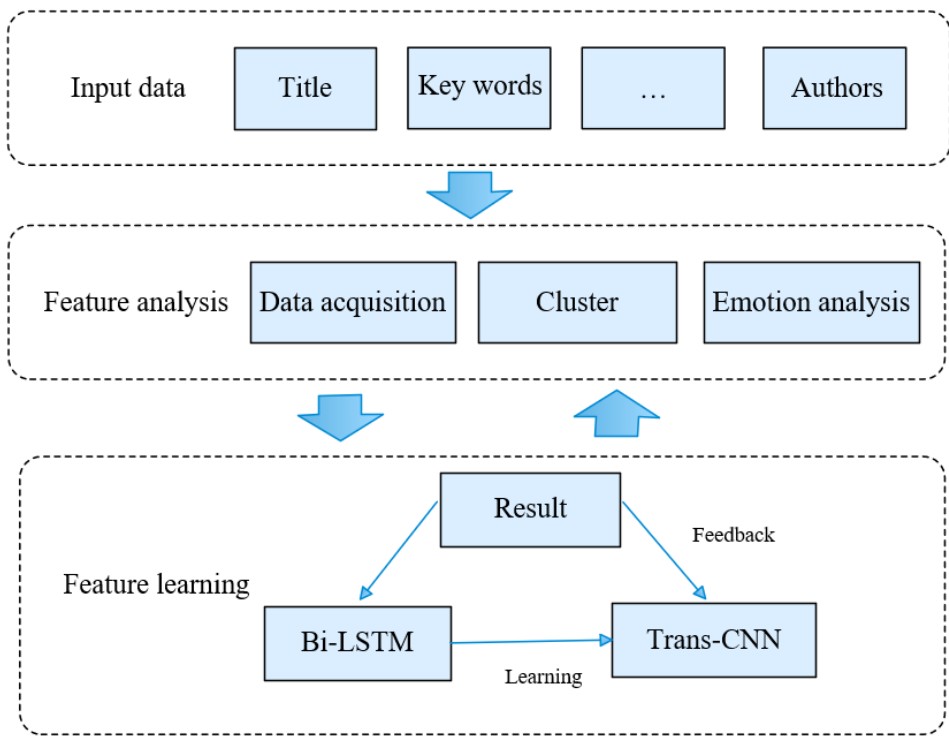

**Figure 1** The framework of the proposed methods.

## News text clustering

Next, the author introduced the process of News Text Clustering. If there is a Q-dimensional space $S^Q$, the finite set in this space can be expressed as $X = \{x_1, x_2, \ldots, x_n\}$. The initialization is randomly divided into k classes, which can be expressed as $C_2, \ldots, C_k$; If clustering object contains n, the clustering center of the i-th class can be expressed as $Z_1, Z_2, \ldots, Z_k, Z_i = \frac{1}{n}\sum_{j=1}^{n} x_j j \in [1, k]$. The ultimate goal of the K-means is to classify n data points into K clusters, where the point is closest to its mean. Its objective function can be expressed as follows.

$$J = \sum_{i=1}^{k} \sum_{j=1}^{n_j} D_{x_j, z_i}^2 \qquad (2)$$

where $D_{x_j, z_i}^2$ represents the Euclidean distance between the j-th text and the i-th cluster center.

The basic process is divided into four steps (*Rashid, Shah & Irtaza, 2020*; *Kadhim & Jassim, 2021*), which are as follows:

(1) Preprocess the raw data from the document.

(2) Sample from the dataset $X$, collect k points, and determine the initial clustering center;

(3) Calculate the distance between each point $x_i$ and the k cluster center $C_i$ based on Eq (2), and classify $x_i$ into $C_i$ with the closest distance;

(4) Determine whether the solution results meet the convergence requirements, if so, complete the iteration; Otherwise, go back to Step 2 and redo the calculation.

## Emotional semantic analysis

In order to analyze the emotional semantics in news text to predict potential hot information in the news, this section designs a new emotional semantic analysis method. $F_i$ (frame) in the emotional semantic structure can be determined by locating the verbs and adjectives in the sentence and matching the frame semantic dictionary. $E_i$ (emotional theme) in the emotional semantic structure is a type of core frame element in terms of semantic role relationship, that is, the person being evaluated or the subject being evaluated. Its syntactic features have a strong correspondence with the dependency syntax structure. Therefore, a method based on dependency syntax rules is used for identification (*Wei, Liu & Wei, 2020*). The matching rules for frames and emotion themes are:

$$LU_{frame}\left[SBV * ATT - head\right]\left[ADVD_{degree}\right][AD - Vn_{negative}]. \tag{3}$$

In Eq. (3), $LU$ is the emotional word in the review text, SBV is the emotional word dominating a subject component, and theme is the emotional theme frame element.

In addition, the single-modal features are used as the target modal features respectively as the input of the query matrix (Q). The calculation process of multimodal feature D is expressed as:

$$D = Concat\left(X_\alpha, X_\beta\right) \in \mathbb{R}^{N \times 2d_h}. \tag{4}$$

The scaling dot product formula in the multi-head attention mechanism calculates the attention of single-modal. The specific calculation process is as follows:

$$H_{D1}^i = D \times W_{D1}^i \tag{5}$$

$$H_{D2}^i = D \times W_{D2}^i \tag{6}$$

$$H_\alpha^i = X_\alpha \times W_Q^i \tag{7}$$

$$A_\alpha^i = softmax\left(\frac{H_\alpha^i \times \left(H_{D1}^i\right)^T}{\sqrt{d_m}}\right) \times H_{D2}^i \tag{8}$$

where, $i$ represents a specific head in the multi-head attention mechanism, $W_{D1}^i$, $W_{D2}^i$ are mapping matrices, and $d_m$ is the scaling factor.

## BIK-LSTM-CNN

In pursuit of discerning noteworthy trends within classified news texts, this section posits the employment of the BiK-LSTM-CNN methodology for the visualization of news text and the extraction of media hot spots. Initially, the author employ the Skip-gram model within Word2vec to delineate the semantic landscape of the data, transforming lexical entities into vectors of fixed dimensions. Subsequently, the BiK-LSTM framework is deployed to encapsulate bidirectional semantic nuances. Ultimately, the features derived from the BiK-LSTM model are amalgamated with the word embedding features, constituting a comprehensive input for a CNN structure. Convolutional operations are applied to this

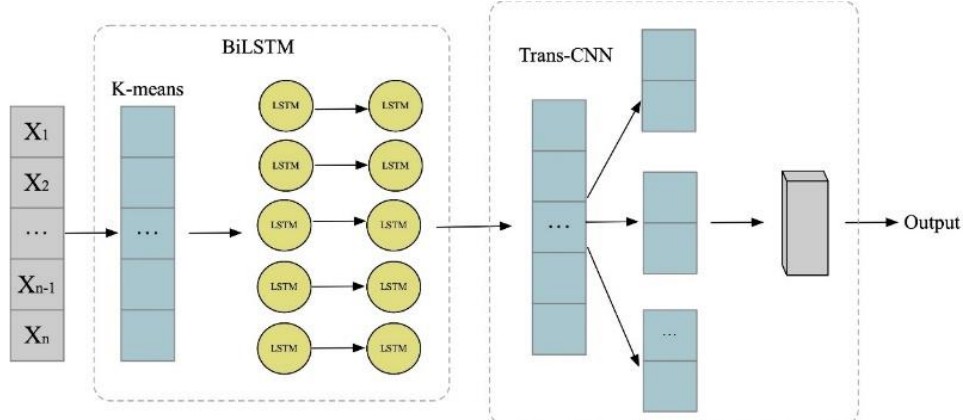

**Figure 2  BiK-LSTM-CNN network structure.**

input utilizing kernels of dimensions 2, 3, and 4. The BiK-LSTM-CNN model, as articulated in this exposition, prominently encompasses a word embedding layer, a BiK-LSTM layer, and a CNN layer. The intricate network architecture is delineated in Fig. 2.

Moreover, the author have refined the architecture of the CNN network for the textual analysis, as delineated in Fig. 3. The devised CNN structure encompasses multiple convolutional layers, max-pooling operations, fully connected layers, and Softmax layers (*Soni, Chouhan & Rathore, 2023*). The fundamental concept underpinning this model involves the utilization of convolution kernels of diverse dimensions to apprehend contiguous N-grams within the textual corpus. Primarily, convolution kernels of varied sizes are employed in the convolutional layer to distill local features intrinsic to the text. The resultant feature vectors undergo a process of down-sampling in the pooling layer, effectively sieving crucial feature information and concomitantly reducing vector dimensions. Subsequently, the feature vectors, having undergone 1-maxpooling, are intricately concatenated, paving the way for multiclass classification through the fully connected layer and the application of the Softmax function (*Liu et al., 2022*). This refined CNN, characterized by its adeptness in capturing proximate and superficial features of text, boasts a streamlined network architecture, thereby expeditiously extracting local textual features.

# EXPERIMENT AND ANALYSIS

## Dataset

Table 1 shows the experimentation, which was conducted on a Windows 10 system, featuring a compute core of 4C+4G, operating at a frequency of 1.8 GHz, and equipped with a 4GB RAM. The computational environment employed Jupyter Notebook alongside Python 3.7. The dataset utilized for this study originates from the amalgamation of THUCNews dataset (DOI 10.5281/zenodo.11046814) and SougouCS dataset (DOI 10.5281/zenodo.5259056). Given the focus on news items predominantly chronicling campus activities, each news piece spans a word count ranging from 400 to 700 words.

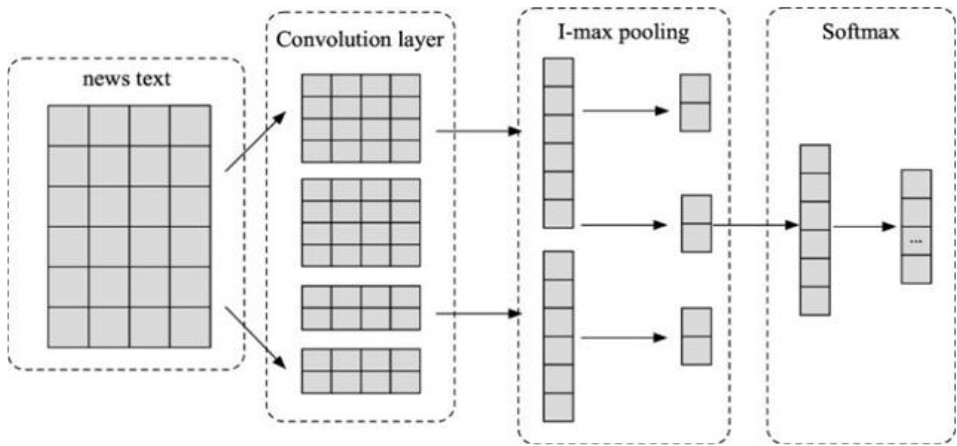

**Figure 3** The CNN network structure.

**Table 1** Experimental environment.

| Items | Introduction |
|---|---|
| Operating System | Windows 10 |
| Processor | 4C+4G, 1.8 GHz |
| RAM | 4GB |
| Software Environment | Jupyter Notebook, Python 3.7 |
| Dataset Source | THUCNews dataset (doi: 10.5281/zenodo.11046814) and SougouCS dataset (doi: 10.5281/zenodo.5259056) |
| Dataset Format | CSV |
| Dataset Size | 11,456 news items |
| News Content | Word count per item: 400–700 words |
| Categories | Education and teaching, graduation and employment, competitions, ideological and political, exchange conferences, learning and training, management services |

Categorically, the 11,456 news items have been classified into 7 distinct categories, namely education and teaching, graduation and employment, competitions, ideological and political, exchange conferences, learning and training, and management services. The compiled dataset is formatted in CSV for ease of accessibility.

This manuscript utilizes the *train_test_split* function from the PyTorch library to partition the dataset, employing the 'stratify' parameter configured to the label column of the dataset. Notably, the proportions of distinct labels within both the training and test sets, derived through this stratified division, mirror the label distribution in the original dataset. To achieve an equitable distribution across categories, each classification is uniformly represented based on the dataset size. The test set is allocated a ratio of 0.2, signifying that 80% of the data is selectively assigned to the training set through random sampling, maintaining the proportional representation of each label from the original dataset.

**Table 2  Transformer model parameter settings.**

| Parameter | Value | Parameter | Value |
|---|---|---|---|
| Number of convolution kernel | 256 | Dropout | 0.5 |
| Convolution kernel size | 2,3,4 | Learning rate | 0.001 |
| Activation function | Relu | Epochs | 6 |
| Loss function | Cross entropy | Batch | 128 |

**Table 3  AdvLSTM model parameter settings.**

| Parameter | Value | Parameter | Value |
|---|---|---|---|
| Hidden node | 300 | Learning rate | 0.001 |
| Epochs | 7 | Optimization function | Adam |
| Batch_size | 128 | Loss function | Cross entropy |

Consequently, the remaining 20% is designated as the test set. The ultimate training set comprises 10,000 items, while the test set encompasses 1,456 items.

### Evaluation indices and parameter setting

The experiments detailed in this manuscript encompass three distinct models, specifically, Transformer (*Gupta et al., 2022*), AdvLSTM (*Yadav, Verma & Katiyar, 2023*), and NewsRNN (*De Souza Pereira Moreira, 2018*). Each model represents a different architectural approach to the problem at hand. Transformer is known for its attention mechanism, which is effective for capturing dependencies between distant words in sequences. AdvLSTM represents a variant of LSTM that may incorporate enhancements such as attention mechanisms or additional gating mechanisms for better memory retention. NewsRNN, focuses on processing sequential data such as news articles, possibly with recurrent connections tailored for this specific type of data.

The parameter configurations for BiK-LSTM-CNN are elucidated in both Tables 2 and 3.

Evaluation of the models' classification efficacy on news text data is gauged through metrics such as macro precision rate (macro-P), macro recall rate(macro-R), and macro F1(macro-F1). as expressed in Eqs. (9), (10) and (11).

$$macro-p = \frac{1}{n}\sum_{i=1}^{n}P_i \tag{9}$$

$$macro-R = \frac{1}{n}\sum_{i=1}^{n}R_i \tag{10}$$

$$macro-F_1 = \frac{2macro-P*marco-R}{macro-P+marco-R} \tag{11}$$

where $P_i$ represents the precision for the $i$-th class, and $R_i$ represents the recall for the $i$-th class, and $n$ is the total number of classes.

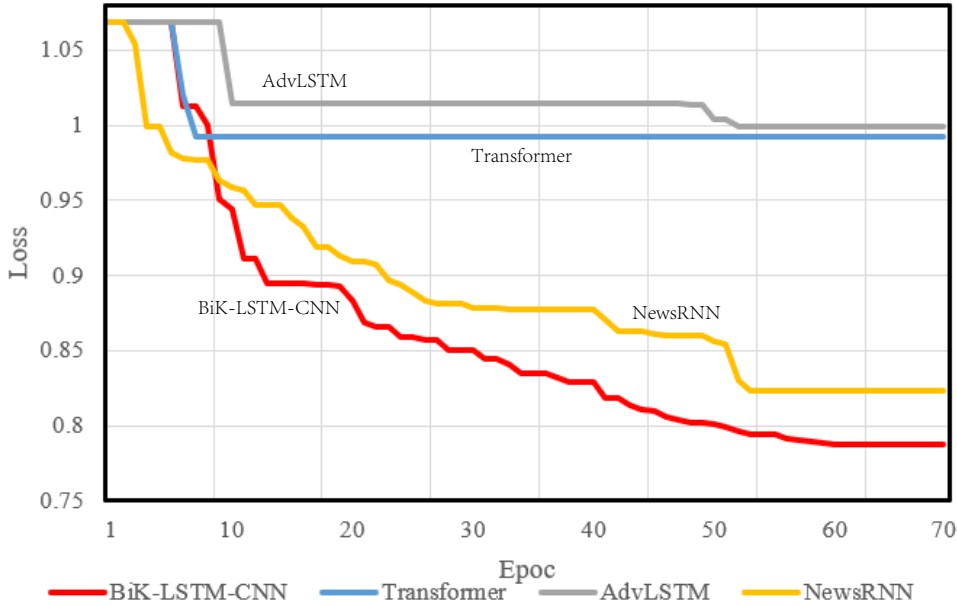

**Figure 4** **Training loss on the THUCNews dataset.**

## Model training process

During model compilation, the chosen loss function is the cross-entropy loss, selected to assess the disparity between the probability distribution derived from current training and the actual distribution. Figures 4 and 5 visually depict a schematic representation of the convergence results, illustrating a comparative analysis between the algorithm proposed in this manuscript and the reference algorithm.

The optimizer parameter is set to 1e−5, and the chosen metric is accuracy. The convergence speed serves as a pivotal metric delineating the algorithm's visual design efficacy. A swifter convergence implies heightened efficiency in attaining optimal values. From the comparative results, it is discernible that the proposed BiK-LSTM-CNN model exhibits relatively low losses on both THUCNews and SougouCS datasets. Nonetheless, there remains a need for improvement in the convergence speed.

## Model comparison

Figures 6 and 7 illustrate notable enhancements on the THUCNews dataset achieved by the BiK-LSTM-CNN model in comparison to Transformer, AdvLSTM, and NewsRNN models. Specifically, the macro precision rate demonstrates a rise of 0.44% and 0.57%, while the macro recall rate sees an increase of 0.46% and 0.69%, respectively. On the SougouCS dataset, BiK-LSTM-CNN outperforms Transformer, AdvLSTM, and NewsRNN models by elevating macro accuracy by 0.50%, 0.91%, and 1.34%, respectively. The macro recall rates experienced increments of 0.51%, 1.24%, and 1.26%. Furthermore, the Macro F1 exhibits improvements of 0.52%, 1.23%, and 1.92%, respectively. The pronounced

PeerJ Computer Science _______________________________

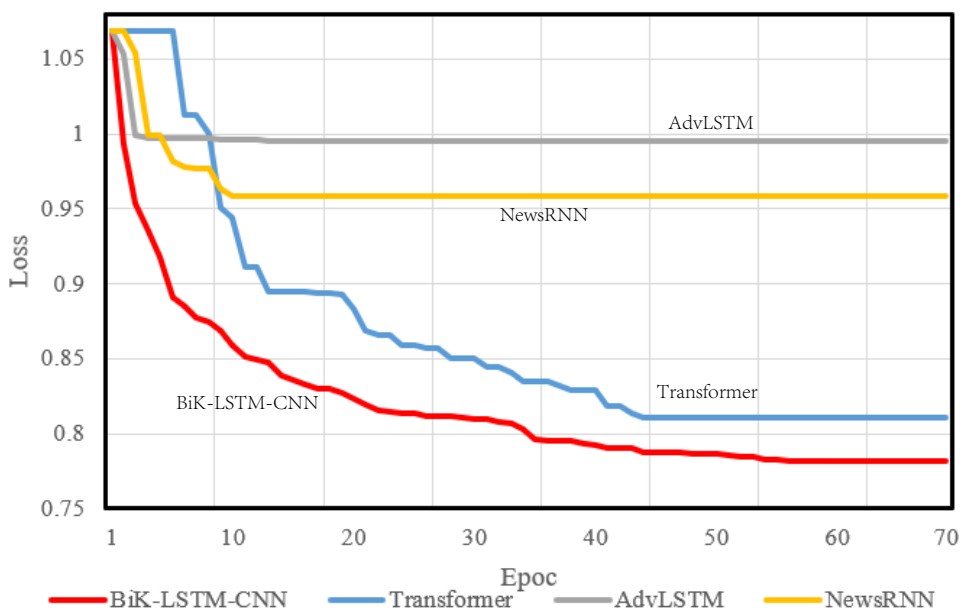

**Figure 5** Training loss on the SougouCS dataset.

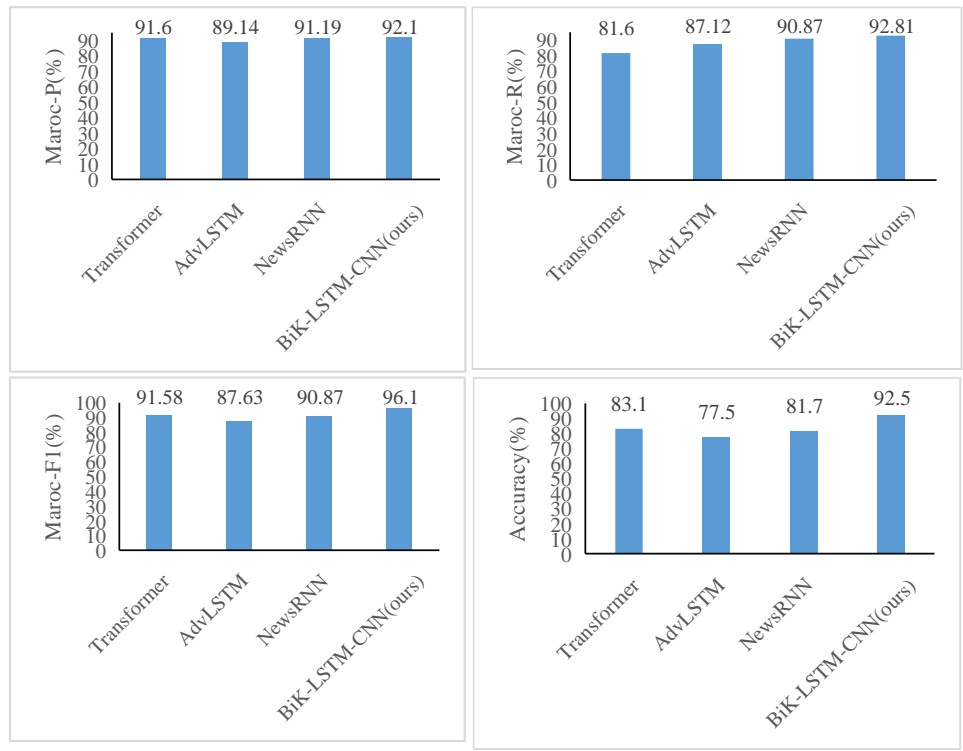

**Figure 6** Experimental comparison results of the SougouCS dataset.

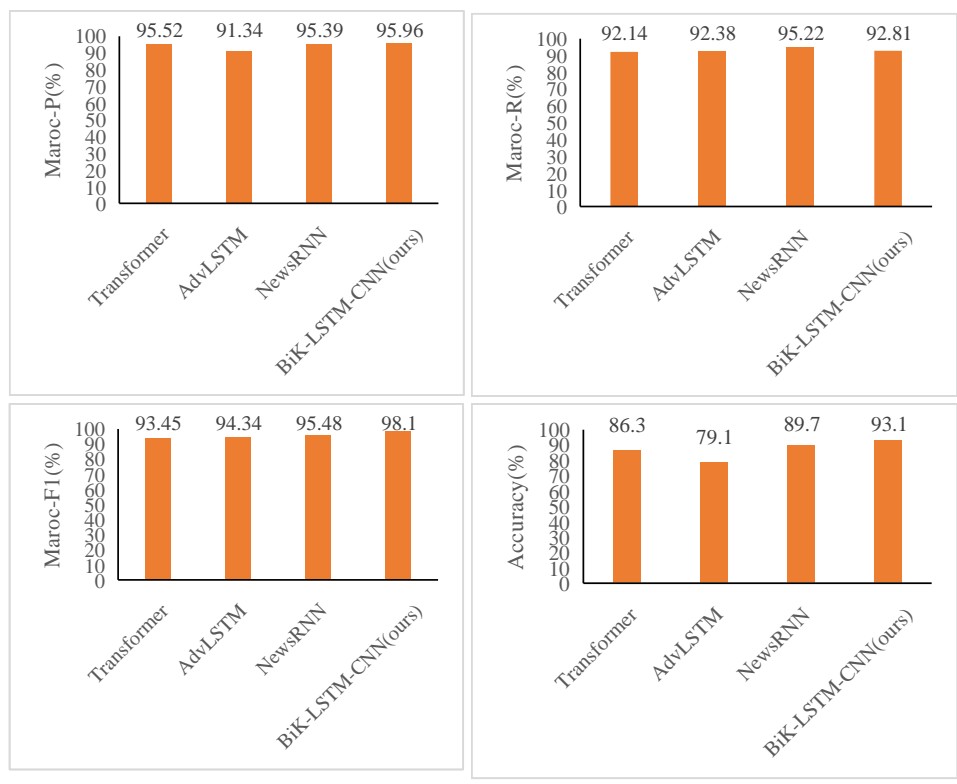

**Figure 7 Experimental comparison results of the THUCNews dataset.**

improvement on the SougouCS dataset compared to the THUCNews dataset is attributed to the imbalanced sample distribution in each category within the THUCNews dataset. Additionally, on the SougouCS dataset, the test accuracy for the NewsRNN, AdvLSTM, and Transformer models is 81.70%, 77.55%, and 78.84%, respectively. Observably, the performance of these four models on the test set is slightly lower than their performance on the training set, suggesting an optimal number of training iterations for each model. Simultaneously, the accuracy of the BiK-LSTM-CNN model surpasses all others, reaching 92.5%, and on the THUCNews dataset, it attains 93.1%. The minimal difference in accuracy between the training and test sets further attests to the well-balanced training level of the model in this study.

Analysis of Fig. 8 reveals that with a sample size of 200, the response time for BiK-LSTM-CNN news hot spot prediction is 17 ms. In contrast, Transformer's hot spot prediction takes 20 ms, and the overall system response time is 15 ms. With a sample size of 300, the response time for Transformer system's news hotspot prediction extends to 23 ms. NewsRNN's news hotspot prediction response time is 27 ms, while the overall system's news hotspot prediction response time remains at 20 ms. Notably, RNN exhibits the lengthiest response time, attributed to the excessive number of parameters in the model, leading to prolonged deduction times.

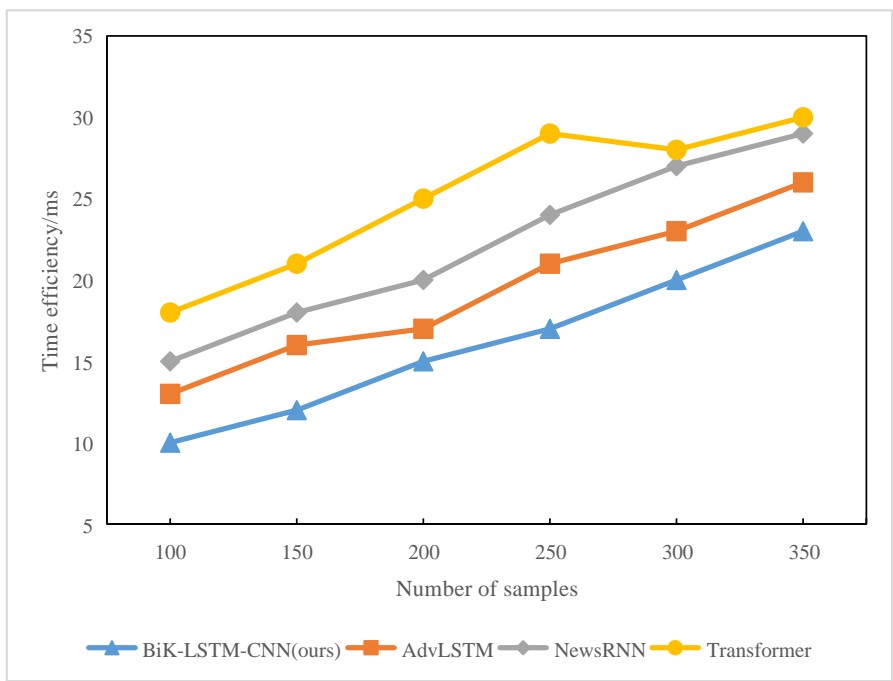

**Figure 8    Time efficiency (ms) of the comparison algorithms.**

## Qualitative analysis and visualization

To showcase the visualization outcomes of BiK-LSTM-CNN for classification, this study categorizes and visually represents sports news, as depicted in Fig. 9. Specifically, four primary categories are discerned, encompassing football, basketball, billiards, and volleyball. The results, as illustrated in Fig. 9, underscore the efficacy of the proposed method in effectively classifying sports news. Notably, basketball news constitutes the most substantial proportion, comprising approximately 30% of the dataset, while billiards news represents the smallest share at around 20%.

Figure 10 illustrates the classification results of feature visualization for BiK-LSTM-CNN. The visualization clearly demonstrates the method proposed in this article's proficiency in classifying distinctive characteristics of various news categories. Notably, entertainment news and sports news exhibit closely aligned characteristics, a phenomenon attributed to the presence of entertainment elements within sports news. The experimental outcomes substantiate that BiK-LSTM-CNN adeptly discerns the intricate features embedded in news text, showcasing its capability to identify deep-seated textual nuances.

## Discussion

The BiK-LSTM-CNN model proposed in this study exhibits promising capabilities for processing and analyzing large-scale text data, particularly in applications requiring efficient classification and prediction tasks. The model's ability to achieve rapid convergence and minimal loss on datasets such as THUCNews and SougouCS underscores its potential in various real-world scenarios. In the domain of online news prediction systems, for instance,

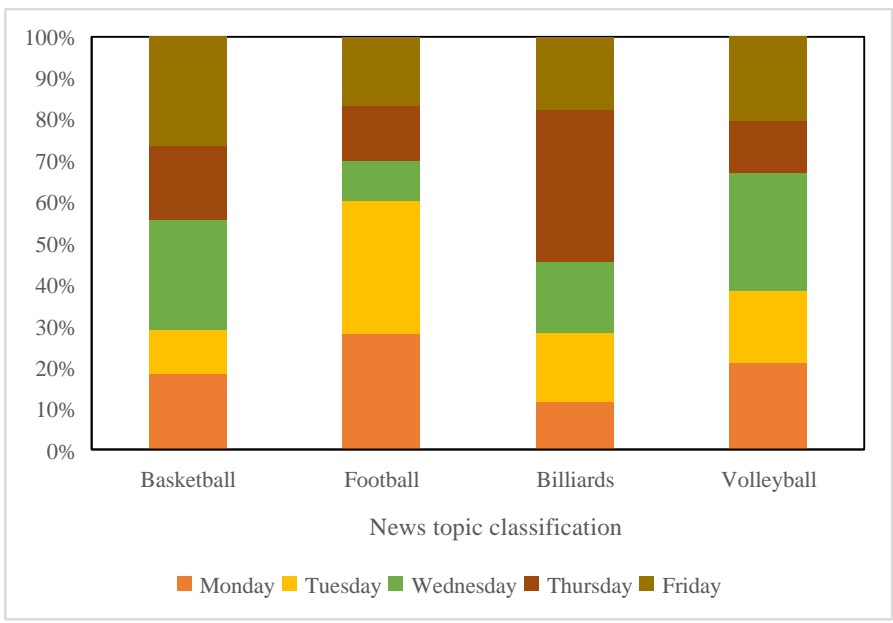

**Figure 9  The tendency of the visualized sports news.**

the incorporation of the Transformer algorithm has significantly reduced response times by eliminating redundant words and optimizing information flow. This enhancement is crucial for applications where timely and accurate prediction of news dissemination popularity is paramount. However, deploying the BiK-LSTM-CNN model in practical settings does pose challenges. One notable issue is the need for continuous optimization to balance recognition accuracy and computational efficiency. Current methodologies, despite advancements in LSTM and CNN architectures, often struggle with extracting deep textual features efficiently. This limitation necessitates ongoing research into hybrid models, such as integrating Transformer networks with recurrent neural networks, to improve both performance and runtime efficiency.

Nonetheless, prevailing methodologies encounter issues such as diminished recognition accuracy and suboptimal time efficiency. The author posits that the primary challenge lies in the incapacity of existing methods to extract profound textual features. Moreover, numerous scholars are dedicated to exploring the construction and enhancement of neural networks. Each standalone neural network model structure inherently carries limitations (*Sarma, 2022*). While scholars strive to optimize models by modifying the information flow within LSTM or by augmenting the depth of convolutional neural networks to enhance classification performance, such endeavors often incur an increase in model runtime (*Wang et al., 2023*). Therefore, amalgamating the transformer network with recurrent neural networks and their variant structures to formulate a hybrid model emerges as a crucial avenue for model optimization. Consequently, this article introduces an enhanced BiK-LSTM-CNN, integrated with emotional semantic analysis, to realize high-dimensional news text visualization and media hot spot mining methods.

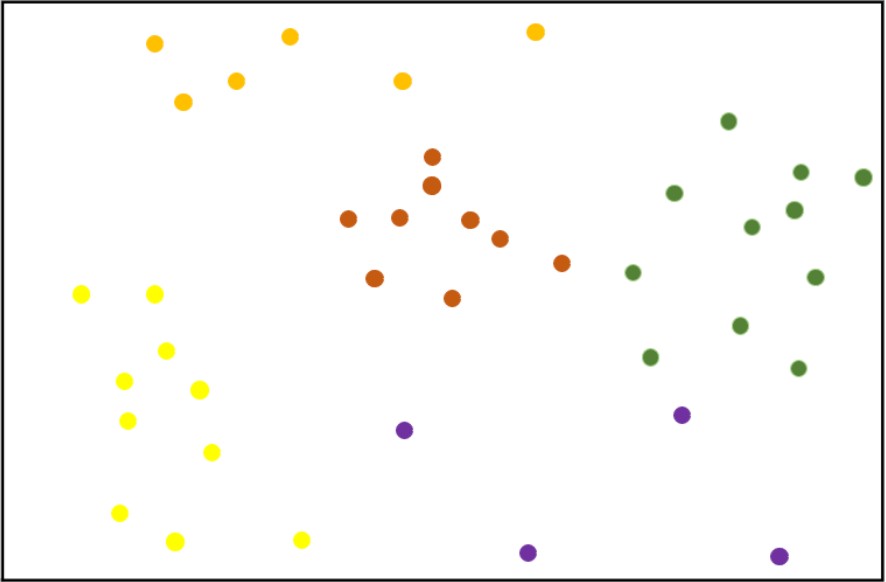

**Figure 10** Visualization of news features.

BiK-LSTM-CNN integrates multiple deep learning models and applies them to news text tasks. First, the pre-trained model LSTM is used to perform high-dimensional mapping on the text data to obtain the embedding vector. Secondly, the feature information of the input text is extracted through the double-layer BiLSTM to obtain the global context information. The deeper key information is extracted through CNN again. Finally, softmax is used to predict the category of the text. The classification accuracy of the BiK-LSTM-CNN algorithm has improved. It is believed that this is due to the fact that BiK-LSTM-CNN retains more semantic and grammatical information in the word vector conversion process.

While the BiK-LSTM-CNN model demonstrates notable strengths in achieving rapid convergence and high classification accuracy, several limitations and areas for improvement warrant discussion. The model shows efficient convergence to optimal values, attributed to the incorporation of the Transformer algorithm. However, further enhancement in convergence speed remains an area for improvement. Future research could explore advanced optimization techniques or parameter tuning strategies to expedite convergence without compromising accuracy. In addition, despite its effectiveness, the model's runtime may increase when integrating complex neural network architectures such as BiLSTM and CNN. This trade-off between model complexity and computational efficiency presents a challenge. Techniques like model pruning or quantization could be investigated to reduce computational overhead while maintaining performance. While effective on the

THUCNews and SougouCS datasets, the model's generalizability to diverse news domains or languages needs validation. Future studies could focus on domain adaptation techniques or multi-task learning frameworks to enhance cross-domain performance.

## CONCLUSIONS

To address the challenges of low accuracy and time inefficiency in existing methods for news text visual extraction and media hot spot mining, this article introduces an enhanced BiK-LSTM-CNN coupled with emotional semantic analysis, aiming to achieve high-dimensional news text visual extraction and media hot spot mining. Experimental outcomes, conducted on the THUCNews and SougouCS datasets, demonstrate that BiK-LSTM-CNN exhibits superior performance compared to CNN, AdvLSTM, and NewsRNN models. The results affirm that the proposed news text visual extraction and media hotspot mining methods can furnish effective data support for news dissemination.

Future endeavors can amalgamate the two research directions of news text visual extraction and media hotspot mining to conduct more comprehensive investigations. For instance, researchers may delve into synergizing news text visual extraction technology and media hotspot mining technology to automatically generate news reports or public opinion analysis reports on media hotspots. Moreover, researchers may investigate methods to enhance the integration of diverse modalities (text, images, videos) within the BiK-LSTM-CNN framework. This could involve developing novel attention mechanisms or fusion strategies that effectively combine textual and visual information for more comprehensive news understanding.

### Funding
This article is the research result of the university-level innovative research team of Hubei University of Science and Technology "Research on the Development of the Meta-universe and Media" (No. 2022T06). The funders had no role in study design, data collection and analysis, decision to publish, or preparation of the manuscript.

### Competing Interests
The author declares that there are no competing interests.

### Author Contributions
- Qingxiang Zeng conceived and designed the experiments, performed the experiments, analyzed the data, performed the computation work, prepared figures and/or tables, authored or reviewed drafts of the article, and approved the final draft.

### Data Availability
The code is available in the Supplementary File.

The THUCNews dataset is available at Zenodo: None. (2024). THUCNews [Data set]. Zenodo. https://doi.org/10.5281/zenodo.11046814.

The SougouCS dataset is available at Zenodo: Zhang. (2021). Sogou News Corpus (SOGOU) (Version v1) [Data set]. Zenodo. https://doi.org/10.5281/zenodo.5259056.

## Supplemental Information

Supplemental information for this article can be found online at http://dx.doi.org/10.7717/peerj-cs.2213#supplemental-information.

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
