# Peer review of "Enhanced analysis of large-scale news text data using the bidirectional-Kmeans-LSTM-CNN model"

_PeerJ Computer Science, doi:10.7717/peerj-cs.2213_

## Round 0.1 · original submission · Major Revisions

Dear contributors
You paper has been reviewed by the experts with great interest and you will see that they have couple of comments for the improvements of your article. I do agree with them and hence advice to prepare a detailed response and update the article in light of the comments and resubmit.

The title needs some rephrasing to make the clarification of the contributions.eg enhanced analysis might not be a suitable word.

Please also improve the language of the manuscript as there needs some structural changes in order for better readability.

Please define all the acronyms at first use.

What is the uniqueness of your article over the others available approaches?

Reviewer 1 ·

Basic reporting

No Comments

Experimental design

No Comments

Validity of the findings

No Comments

Additional comments

Addressing the challenges of low accuracy and time efficiency in existing methods for news text visual extraction and media hot topic mining, this paper introduces an enhanced BiK-LSTM-CNN model combined with sentiment semantic analysis. The experimental results conducted on the THUCNews and SougouCS datasets demonstrate that the BiK-LSTM-CNN model can provide effective data support for news dissemination. However, there are still some issues in the scheme description and experimental process that need to be improved.

(1) In Section 2.2 of the text classification model module, the author summarized related work that was conducted prior to 2020. To highlight the novelty of the author's work, please include a summary of relevant research conducted since 2023.

(2) Some symbols and parameters are not defined in the text. Please provide explanations for all the parameters mentioned throughout the article to enhance readability.

(3) For the text clustering process described in Section 3.3, a corresponding flowchart can be created to improve readability.

(4) The English writing should be polished regarding grammatical, lexical, and punctuational points. Avoid colloquial expressions and ensure the language is formal and academic.

(5) The author should include a brief description at the beginning of each subsection in the methodology section to enhance the connection with the preceding text.

(6) In the experimental analysis, this paper selects the methods from literatures [33], [34], and [35] as comparison schemes. However, the author has not provided an analysis or explanation of these schemes in the related work section. Therefore, the author should clarify the rationale for selecting these three comparison schemes in Section 4.2.

(7) An overview of potential future research directions could include exploring improvements to the BiK-LSTM-CNN model to achieve multimodal news information extraction and hot topic mining.

·

Basic reporting

No Comments

Experimental design

No Comments

Validity of the findings

No Comments

Additional comments

Focusing on the issues of low accuracy and time efficiency in the existing methods for identifying and clustering large amounts of data, and starting from capturing the subtle emotional and sentimental backgrounds in news texts, this article proposes an improved Bidirectional-Kmeans-Long Short-Term Memory Network-Convolutional Neural Network (BiK-LSTM-CNN) model. This article has a certain degree of innovation, but further modifications are needed.
(1) In the introduction, the author argues that traditional text processing methods are unable to effectively identify key information and emerging hot topics. Please analyze the reasons for the inefficiency of traditional text processing methods in the article.

(2) The BiK-LSTM-CNN model designed by the author comprises four modules: news text preprocessing, news text clustering, sentiment semantic analysis, and the core BiK-LSTM-CNN model. Please elaborate on how these four modules interact to achieve comprehensive data analysis and inspection.

(3) Current methods for visually extracting information and mining from texts are primarily divided into traditional methods and deep learning-based methods. The author can describe and summarize related work according to traditional methods and deep learning-based methods in the relevant sections.

(4) In Section 3.4, focusing on sentiment semantic analysis in news texts, the author proposes a novel sentiment semantic analysis method. Please enhance the description of the proposed method in this section. For instance, what operation does each formula from (5) to (8) perform?

(5) The author can present the experimental environment and other relevant data in Section 4.1 in a tabular format and introduce the selected datasets. Additionally, the author should clarify whether data cleaning or other preprocessing steps were necessary before the experiments, to facilitate reproducibility for readers.

(6) It is common for new methods to have limitations or areas for improvement. Therefore, it would be valuable for the authors to discuss any potential shortcomings of their proposed method and recommend techniques to overcome these limitations in the revised manuscript. This would help readers gain a better understanding of the proposed method's strengths and weaknesses and how it compares to existing methods. Additionally, discussing potential limitations and ways to address them can help to guide future research and development efforts in the field.

Reviewer 3 ·

Basic reporting

Introduction provides a good overview of the challenges in traditional text mining methods, it could benefit from a clearer statement of the specific goals and contributions of the paper. Clearly articulating the gap in existing methods that the proposed BiK-LSTM-CNN model aims to address would help readers understand the significance of the research.
Please provide more in-depth explanations, such as the specific techniques used for news text preprocessing, clustering, and sentiment analysis, would enhance the reader's understanding of the model's inner workings and its potential applicability to other text mining tasks.
A detailed discussion on the dataset(s) used, including their size, diversity, and any preprocessing steps applied, would enhance the reproducibility and generalizability of the findings. Additionally, a more thorough analysis of the results, including statistical significance tests and potential limitations of the approach, would strengthen the paper's conclusions.
The paper mentions the potential applications of the BiK-LSTM-CNN model in processing and analyzing large-scale text data, but could further elaborate on the practical implications of its findings. Discussing specific real-world scenarios or industries where the proposed model could be applied, along with any potential challenges or limitations in deployment, would enhance the relevance and impact of the research.
Finally, while the overall language and clarity of the paper are good, there are some instances where the writing could be further refined for clarity and precision. Reviewing the text for grammatical errors, awkward phrasing, or ambiguous statements would help ensure that the message is conveyed effectively to the reader.
By addressing these improvement areas, the paper could further enhance its contribution to the field of text mining and increase its potential impact on both research and practical applications.

Experimental design

As above

Validity of the findings

As above

Additional comments

As above

---

## Round 0.2 · accepted · Accept

Thanks for your submission, I'm pleased to inform you that reviewers are satisfied with the current version of the paper. Congratulations

Reviewer 1 ·

Basic reporting

All communicated modifications have been incorporated in the revised manuscript. Paper is accepted as it is.

Experimental design

All suggested modifications have been incorporated. Now it's OK.

Validity of the findings

OK

Additional comments

Paper is accepted as it is.

·

Basic reporting

It is worthy to be published.

Experimental design

Design is good and recommended for acceptance.

Validity of the findings

All the code is valid.

Additional comments

All modifications has been incorporated

Reviewer 3 ·

Basic reporting

In this study, the authors have done what was requested in the revision process. The article has become scientifically adequate in this form. It is appropriate to accept the manuscript in this form.

Experimental design

In this study, the authors have done what was requested in the revision process. The article has become scientifically adequate in this form. It is appropriate to accept the manuscript in this form.

Validity of the findings

In this study, the authors have done what was requested in the revision process. The article has become scientifically adequate in this form. It is appropriate to accept the manuscript in this form.

Additional comments

In this study, the authors have done what was requested in the revision process. The article has become scientifically adequate in this form. It is appropriate to accept the manuscript in this form.